# Microphysiological Systems for Neurodegenerative Diseases in Central Nervous System

**DOI:** 10.3390/mi11090855

**Published:** 2020-09-16

**Authors:** Mihyeon Bae, Hee-Gyeong Yi, Jinah Jang, Dong-Woo Cho

**Affiliations:** 1Department of Mechanical Engineering, Pohang University of Science and Technology (POSTECH), 77 Chungam-ro, Nam-gu, Pohang 37673, Korea; bmh0627@postech.ac.kr; 2Department of Rural and Biosystems Engineering, College of Agricultural Sciences, Chonnam National University, 77 Yongbong-ro, Buk-gu, Gwangju 61186, Korea; 3Department of Creative IT Engineering, Pohang University of Science and Technology (POSTECH), 77 Chungam-ro, Nam-gu, Pohang 37673, Korea; 4Institute of Convergence Science, Yonsei University, 50 Yonsei-ro, Seodaemun-gu, Seoul 03722, Korea

**Keywords:** neural microphysiological system, neurodegenerative disease, neural cell, extracellular matrix, organ-on-a-chip, 3D cell-printing

## Abstract

Neurodegenerative diseases are among the most severe problems in aging societies. Various conventional experimental models, including 2D and animal models, have been used to investigate the pathogenesis of (and therapeutic mechanisms for) neurodegenerative diseases. However, the physiological gap between humans and the current models remains a hurdle to determining the complexity of an irreversible dysfunction in a neurodegenerative disease. Therefore, preclinical research requires advanced experimental models, i.e., those more physiologically relevant to the native nervous system, to bridge the gap between preclinical stages and patients. The neural microphysiological system (neural MPS) has emerged as an approach to summarizing the anatomical, biochemical, and pathological physiology of the nervous system for investigation of neurodegenerative diseases. This review introduces the components (such as cells and materials) and fabrication methods for designing a neural MPS. Moreover, the review discusses future perspectives for improving the physiological relevance to native neural systems.

## 1. Introduction

Neurodegenerative diseases (NDs) are characterized by progressive neurodegeneration with aging, and destruction of the main functions of the central nervous system. They generally cause problems in cognitive function, motor function, and the maintenance of homeostasis. These diseases have become more severe with the rapid progression of aging societies. In 2019, the World Health Organization (WHO) announced that the number of dementia patients reached approximately fifty million, with ten million new patients every year. However, there are not enough therapeutic options for NDs, as most approaches merely address symptom alleviation. Although many medications are listed as candidates in pipelines for drug development, most recent drugs have failed in the final clinical stages, owing to insufficient performance [1,2]. A primary challenge in drug development for ND includes consideration of the various pathophysiologies of NDs and the complexity of the central nervous system (CNS) [3]. Recent failed clinical trials targeted only a single pathology, and/or the research models insufficiently reflected the physiology of the human CNS. Thus, an understanding of the multifactorial physiology of an ND is required to develop reliable medications.

NDs exhibit extreme complexity, owing to the combination of complex anatomical physiology, diverse external factors, and various pathologies. The CNS has specialized features, including highly compartmentalized neural networks, in addition to a complex extracellular matrix (ECM), diverse cellular population, and blood-brain-barrier (BBB), all of which are highly involved in the function of the CNS. In the case of Alzheimer’s Disease (AD), which is one of the representative neurodegenerative disease, a few molecules (such as amyloid-beta accumulation and pathological tau formation) have been revealed as responsible for destroying the CNS and inducing AD progression; however, a large part of the solution still remains unclear in the search for a drug target to treat the irreversible destruction from AD [4]. Besides, NDs have different onset times and pathologies for each patient, depending on genetic and environmental factors [5]. Even though the understanding of ND pathology has progressed significantly, the complexities of serial and sporadic destruction events have not yet been fully understood from current research models. The lack of understanding regarding neurodegeneration is the primary hurdle in searching for a hit compound and/or an advanced drug solution for ND. Therefore, a more physiologically relevant research model is needed to investigate the underlying mechanisms in neurodegeneration by capturing the anatomical features of the CNS, external factors, and physiological changes, as in a human CNS.

Animal models have been widely used as research models for ND. The models can reproduce neurodegeneration in a specific brain region to investigate the symptoms of ND. However, the models are restricted to determining the overall pathological hallmarks of ND. For example, a 5X Familial Alzheimer’s Disease (FAD) mouse was established through genetic modification to overexpress AD-linked genes and secret, a type of amyloid peptide that increases intracranial toxicity and induces neuronal damage [6,7]. This model has contributed to the investigation of the amyloid pathology of AD; however, the mouse could not reproduce pathologies other than amyloid accumulation. Likewise, genetic engineering approaches, such as overexpression, gene insertion, and knock-in/out, are beneficial to investigating the progression of familial disease; however, they are limited in capturing other pathologies that sporadically originate, i.e., that are not from a genetic disorder. Moreover, animals are inherently different from humans, resulting in many discrepancies in anatomy, disease development mechanisms, drug metabolism, and pharmacokinetics [8]. In summary, a supplementary research model is required to overcome the translational gaps with regard to the pathology and other factors between humans and animals.

In vitro models have been used as a research models for ND. The 2D cell culture method, a traditional in vitro research model, has contributed to the study of the fundamental pathologies of NDs based on a simplified environment [9,10]. However, the models cannot fully consider the dynamic nature and 3D physiological features of the native CNS, such as the BBB, neural network, and compartmentalized neural layer. To overcome this limitation, 3D culture models have been developed for the CNS. Among the 3D culture models, the neural microphysiological system (neural MPS) is used for neural tissues, and is uniquely engineered to recreate the physiological interactions in the CNSs of humans. The system is defined as a research device to capture the human-like physiologies of tissues and organs in vitro, i.e., the structures and physiochemical factors of the tissues [11,12,13,14,15,16,17]. The physiological structures of the neural MPS are constructed with the proper cell sources, materials, and fabrication methods, to mimic the interactions in the interfaces of the neural tissue. The neural MPS has advantages with regard to adopting various cell sources, materials, and fabrication methods to reproduce multiple physiological environments, including physical cues (e.g., microfluidic cues, electric stimuli). In addition, the neural MPS can favorably induce pathological factors with human-derived cell sources to determine the pathological correlations between NDs and the CNS. Based on its potential, the neural MPS has emerged as an approach to investigating NDs, e.g., to determine human-reliable pathological factors.

This review aims to introduce the essential elements (particularly the cell sources and materials) in the design of a neural MPS for studying NDs. In addition, we explain the current fabrication technology for neural MPS design. First, we address how neural cells reproduce the functionality of the brain and the pathology of NDs, to recreate the physiological environment of the brain. We introduce fabrication methods to build a structure of the brain with examples of neural MPS. We also discuss the application of neural MPS to the study of NDs, especially with respect to Alzheimer’s disease because many examples of neural MPSs have highly contributed to the research on Alzheimer’s disease (Figure 1). Furthermore, the review discusses possibilities for improving the neural MPS, to better capture the complexities of NDs and the CNS.

## 2. Neural Cells for Neural Microphysiological System (Neural MPS) Design

Proper neural cell sources are critical to reproducing the pathophysiological environment for the CNS. We cover recently used neurons and types of glial cells to design a neural MPS in the context of the pathophysiology for an ND.

### 2.1. Neurons

The brain is composed of various types of neurons. The neurons form a neural network in the brain, to propagate cellular signals between themselves [18,19,20]. The transmissions of signals between neurons are involved in performing certain functions of the nervous system, including regulating voluntary movements, stress, and homeostasis in the body. As a neural network is destroyed owing to abnormalities in the neurons, signal transmission between the cells becomes less smooth, causing problems with the functions mentioned above. In the case of AD, toxic amyloid-beta plaques cause apoptosis in nerve cells or pathological tau proteins accumulate in degenerating neurons, inducing the axons to collapse, and ultimately causing cognitive impairment. Also in the case of Parkinson’s disease (PD), dopamine levels decrease since dopaminergic neurons in the substantia nigra gradually break down owing to abnormal lewy bodies in the stem [21,22,23]. As a results, PD induces dysfunction to motor ability, such as tremor, rigid muscles, loss of automatic movements, and sometimes melancholy. As such, the primary pathology of NDs destroys the functions of neurons, and sequentially induces functions in other neuronal cells. Below, we cover a variety of neural cells that reproduce mature neurons; these have been recently used in studies on pathophysiology.

#### 2.1.1. Human-Induced Pluripotent Stem Cell-Derived Neuron

The induced pluripotent stem cell (iPSC) has been raised for neurological disease modeling, and has the potential to differentiate into several types of cells [24,25,26]. Owing to the pluripotency of the cells, iPSCs can differentiate into neural cells, including neural progenitor cells, astrocytes, and even pericytes. Of particular note, iPSCs can represent the patient-specific genetic diversity in each patient source. For example, early-onset AD has mutated gene conditions known as *PSEN1*, *PSEN2,* and Amyloid precursor protein; these express the pathophysiologies of the amyloid-beta plaque and pathological tau formations [27,28,29]. Furthermore, sporadic AD has a poorly understood genetic phenotype relative to FAD. Human iPSC-derived neural cells are favorable for determining these unrevealed risky genes without exogenous gene modification [30]. Moreover, iPSC-derived neurons show not only genetic phenotypes but also neuropathology, e.g., elevated Aβ levels in patient-derived cells relative to those in healthy cell sources [31,32,33].

However, iPSC-derived neurons are in the relatively early stages, and may not express enough of an aging factor for ND study. The expression of an aging factor in an iPSC is required to enhance the degeneration conditions for neural tissue. Another limitation is that iPSC-derived neural cells have an inconsistent cell type population. The limitation makes iPSC-derived neurons difficult to analyze in unified experimental conditions. Nonetheless, iPSC-derived neurons have been widely used in designing a 3D brain-on-a-chip models [34,35,36,37], neurotoxicity testing [38], and disease modeling [39,40], owing to its advantages for various cellular populations and genetic conditions (Figure 2). These models are expected to be used to develop personalized medicines (notably, for sporadic ND patients).

#### 2.1.2. Human Fetal Tissue-Derived Neuron

A human fetal brain-derived neuron is an evident cell source for reviewing the phenotypes of neurons for in vitro models. However, the primary neural progenitor cell from the fetal brain is limited with regard to proliferation, restricting the usage of the cells in reproducing microphysiological system designs. To overcome this limitation, immortalized neural progenitor cells have been used; these can proliferate over dozens of passages. For example, an ReN cell is a commercialized neural progenitor cell derived from the cortex or ventral mesencephalon region of human fetal brain tissue. After infection with retrovirus to induce immortalization, the cell can act as one of the most potent cell sources for mimicking neural tissue (especially brain tissue), while providing the robust characteristics of neural progenitor cells.

Furthermore, the immortalized neural progenitor cells can be used for ND modeling. Choi et al. [41], infected an ReN cell to express the pathophysiology of AD. The AD gene-infected ReN cells differentiated into neurons and astrocytes, and showed the primary pathophysiology of ND. Based on the research results, the immortalized neural progenitor cell has become a promising cell source for ND modeling. However, the cell source is limited to reproducing the various genetic conditions of each patient, owing to the single origin of the cell. Moreover, the differentiated neurons are relatively younger than the patient’s neurons, and thus may be limited in reproducing degeneration from aging factors. Despite the limitations, human fetal tissue-derived neurons are expected to be used to build feasible in vitro neural tissue, while providing the substantial characteristics of neural cells.

### 2.2. Glial Cells

Glial cells primarily maintain the plasticity of the neural network, and support biochemical transportation in the brain [42]. Recently, the roles of glial cells in NDs have been emerging in the development of a pharmaceutical study for NDs, as the gliosis of astrocytes and activation of microglia have been revealed as playing crucial parts in the pathophysiology of NDs [43]. Below, we introduce an overview of glial cells (which play a significant role in the pathophysiology of NDs), and discuss the types of cell sources for building a physiological neural MPS.

#### 2.2.1. Astrocyte

Astrocytes are the most abundant type of glial cells in the brain. They are the primary homeostatic regulators of the CNS by regulating blood flow in the BBB, delivering nutrients and metabolites in the brain, and controlling neuronal plasticity [44,45,46,47,48]. Additionally, astrocytes react sensitively to pathological triggers, such as trauma, infection, stroke, or NDs. The reaction is defined as a reactive gliosis. Reactive astrocytes change their morphology from a star-like shape to an expanded shape. In addition to morphological changes, the upregulation of the glial fibrillary acidic protein (GFAP) is another hallmark of reactive gliosis. With these molecular and cellular changes, reactive astrocytes help to repair scars in the CNS, restore homeostasis, and/or modulate neurogenesis and neuroinflammation [48,49].

These hallmarks and phenomena of reactive astrocytes can be pathological signs of ND. The reactive gliosis in an ND includes upregulation of the GFAP level, recruitment of microglia, or loosening the tight junctions of the BBB (Figure 3A); this occurs even without external pathological factors, such as injury or trauma. As part of the astrocytes are in “overreaction”, the over-reactive astrocytes induce dysfunction in other glial cells, destruction of neuronal plasticity with excessive glutamate modulation, or immoderate inflammatory responses in the microglia. Therefore, understanding maladaptive gliosis, as well as the occurrence of reactive gliosis, can be a therapeutic target for NDs.

Astrocytes have also been applied to model neural MPSs, e.g., to investigate the physiological relevance between NDs and gliosis. For example, microfluidic-based gliosis models have been used to study the astrocytic network in the CNS [50,51]. Ahn et al. [50] reviewed the BBB structure, which includes a 3D astrocytic network. Astrocytes in the perivascular space participate in regulating water homeostasis via the water channel protein aquaporin-4 (AQP4) [52,53]. Therefore, the localization and polarization of AQP4 on the BBB surface with a 3D astrocytic network is essential to reproducing the physiological functions of astrocytes in the BBB. The study (Ahn et al.) investigated the polarized expression of AQP4 from astrocytes in their engineered BBB model. The model could mimic the water transport system in the BBB for regulating homeostasis. Furthermore, they determined the drug transportation pathways in their system, to investigate its potential in therapeutic models. Hence, the engineered BBB model allowed us to investigate cellular responses regarding homeostasis between the astrocytes and vascular cells in the BBB. Moreover, it is expected to be applied in the therapeutic study of NDs.

There are several technical considerations when applying astrocytes to model the neural MPS more physiologically. First, the reproduction the physiology of astrocytes in vitro (as opposed to in vivo) is an important technical issue, i.e., to adapt astrocytes to in vitro modeling. As astrocytes easily display reactive morphology in physiologically different in vitro environments, it is significant to achieve a physiologically relevant microenvironment in vitro for the in-vivo-like physiology. Therefore, several studies have recently attempted to build microphysiological environments with different materials, physical properties, or dimensions [54,55,56,57]. In summary, the choice of materials for the astrocytes’ culture environment should be carefully considered so as to achieve the in-vivo-like physiology.

Another issue concerns determining the physiological evidence for the difference between the primary astrocyte and cell line. Galland et al. [45] compared the functional and molecular characteristics between primary astrocytes and cell lines. They determined that the primary astrocyte showed a higher expression of the gap junction, glucose metabolism, and astrocytic profile than the cell line, but the reproducibility was better in the cell line. As the cell sources have pros and cons, the choice of cell sources should be carefully considered, so as to build a physiologically relevant environment with the native CNS and provide repeatability for therapeutic study.

#### 2.2.2. Microglia

The brain has a specific immune function based on microglia. Microglia are the primary immune cells in the brain that participate in homeostasis in the CNS. Microglia can be activated by various pathological factors, such as infection, stroke, and neurodegeneration. As microglia are activated, they begin proliferation, and change their morphology from a ramified shape into an amoeboid shape. The activated microglia begin to remove damaged neurons, unnecessary synapses, and infected foreign materials [58,59]. However, microglia have also multifaceted effects, e.g., as a pro-inflammatory response with an M1 phenotype, or an anti-inflammatory response with an M2 phenotype. In a healthy brain, the pro-inflammatory response is at a basal level, to help to clear out necrotic neurons. However, in ND, microglia are chronically activated, and secrete excessive cytokines that damage neurons (Figure 3B) [60,61].

As such, determining the correlation between excessive microglia activation and progression of ND can be a therapeutic goal. Recently, various neural MPS platforms have been suggested for investigating therapeutic methods to address the excessive neuroinflammation in NDs. For example, Park et al. [62] reproduced the activation of microglial inflammation in a physiologically human-like AD environment by employing immortalized human microglia. Notably, they induced microglial migrations, morphology changes, and secretions of pro-inflammatory factors as their AD mimicked the environment. Moreover, they determined that the recruited microglia induced damage to neurons and astrocytes in vitro. The microglia co-culture platform was able to support the pathological mechanisms of microglia in AD with regard to pro-inflammatory cytokine secretion.

To reproduce physiologically relevant microglia behaviors for NDs in vitro, it is essential to mimic and establish “resting” microglia, as in the native brain. For that, a recapitulation of brain microenvironments is significant for “educating” the microglia [63]. Microglia show quantitatively and qualitatively different inflammatory responses in vivo and in vivo. The in vitro environments are relatively more restrictive for educating microglia than the in vivo responses regarding physical cues, biochemical compositions, and pathological factors. Microglia in vitro show an unstable resting state, as they are not accustomed to homeostatic factors in the native brain. As such, there is a need to enhance the extracellular factors learned from cell-cell interactions with other neural cells. Furthermore, it is also important to provide physicochemical properties similar to those in the native brain.

The characteristics of the cell source (e.g., primary, immortalized, or stem cell-derived) should be investigated to establish the resting state of the microglia. For example, the immortalized microglia from a neonatal or embryonic CNS have different phenotypes from elderly microglia, although the proliferation is easier than in other sources. Moreover, stem-cell derived microglia can even reproduce the elderly phenotypes for aged microglia; however, it is hard to stably differentiate the stem cells into microglia. The difficulty can be an obstacle to determining an even resting state of the microglia. Additionally, primary microglia are mainly isolated from animals such as mice; therefore, there is a pathological and genetic gap between the respective primary microglia from animal and human phenotypes. In addition, although the human primary microglia can reproduce human genetic variability, they are restricted to donations alone, as well as to control the antemortem conditions. Therefore, advanced cell culture methods are required to supplement the pros and cons of each cell source, and to achieve ideal resting microglia.

## 3. Materials for Neural MPS Design

Neural cells need an ECM to differentiate, proliferate, and migrate, so as to maintain homeostasis and signal propagation in the CNS [64,65,66]. A physiologically CNS-relevant neural MPS can enhance the correlation between the progression of ND and brain dysfunction. Additionally, functional materials, such as conducting materials, can enhance the neural cell behavior and help to investigate neural cell signal transduction. In this review, we introduce the materials for the artificial ECM building the scaffold or functional recording/sensing systems for the neural cell culture.

### 3.1. Synthetic Biomaterials

Synthetic biomaterials have been applied in tissue engineering to build artificial ECMs. Synthetic materials have advantages with regard to batch-to-batch reproducibility and cost, relative to natural source derived biomaterials. Furthermore, the materials can be tuned to control physicochemical properties such as the modulus, porosity, and protein component ratio in the material [67,68]. In this part, we introduce the polymer types of synthetic biomaterials and their advantages. In addition, we present how the neural MPS adapts the materials in its design, and perspectives for addressing limitations.

#### 3.1.1. Scaffold Materials

To culture neural cells in vitro, an adequate artificial ECM is necessary to support neuron attachment as well as to transport nutrients and metabolites. Neural cells in vitro require a special ECM that is physically as soft as the brain and has a stable structure (for long-term cultures) to incorporate differentiation and cell aging. Furthermore, the ECM needs to provide biochemical binding sites for extending the axon, along with the ECM. In this section, we introduce polyethylene glycol (PEG)-based hydrogel as a biomaterial for building a scaffold to culture neural cells.

-Polyethylene glycol (PEG)-based hydrogel

Polyethylene glycol (PEG) is a hydrophilic polymer which can be activated using various end groups, such as acrylate, vinyl ether, and n-hydroxysuccinimide ester. The mechanical properties, including the degradability, stiffness, and viscosity, can be controlled with different end groups. A PEG hydrogel is a highly cross-linked hydrogel with a porous structure. The porous structure is favorable for providing adequate access to nutrition and metabolite delivery for cell survival. Furthermore, the material has only a slight immune reaction to the cells. Owing to these biocompatible properties, PEG hydrogels are widely used in in vitro cell and tissue cultures [69,70,71].

With particular regard to neural tissue cultures, the material can be used to investigate neural cell behaviors, along with different physical environments. Lampe et al. [72] determined the impacts of using a degradable PEG hydrogel on neural cell behaviors. They copolymerized PEG dimethacrylate and poly-(lactic acid) dimethacrylate to control the degradability of the material. They evaluated how the degradability of the material affects neural cell proliferation and viability. The results indicated that the neurons had improved cellular viability with the more degradable hydrogel, and that the hydrogel could control neural cell behaviors with various degradability values.

Furthermore, a PEG material can be used to test a drug delivery to neural cells. Burdick et al. [70] loaded neurotrophin-3 in a poly-L-lactic acid (PLA)-PEG hydrogel. Then, they investigated the releasing profiles and effects on neural cell outgrowth. The results indicated that the PLA-PEG hydrogel could be applied as a growth factor delivery material, owing to its controllable degradability. Thus, PEG-based hydrogels are favorable for tuning physical properties, especially degradability. However, the typical limitation of PEG-based hydrogels [73] is that the they are not tissue-specific, nor are they adhesive (for cell attachment). As such, natural ECM proteins, such as collagen, fibronectin, and laminin, have recently been added to PEG hydrogels to provide a cell-adhesive environment. The PEG-based hydrogel is expected to provide an alternative strategy when combined with these tissue-favorable proteins.

#### 3.1.2. Conducting Materials

Neural cell behaviors, including differentiation and proliferation, can be controlled with exogenous stimuli, e.g., fluidic, gravitational, and electric stimuli. Physical actuators have been introduced to neural MPSs, to thereby assign adjustable stimulations [74,75,76,77]. One actuator was based on electrical signal transduction to the neuron. Such systems are used to actuate neurons via electrical stimulation. The stimulation not only enhances neural cell behaviors but also treats neurodegenerative diseases, such as Parkinson’s disease and schizophrenia, by deep brain stimulation (DBS). In this regard, in vitro research is required to investigate the clinical treatment effects of electric stimulation on neurons.

Notably, the choice of conducting material is gaining attention in neural tissue engineering with electrical stimuli Metal is rarely considered as a proper material for culturing neural cells as the cells hardly adhere to the metal surface, and the stiffness of the metal is significantly different from native neural tissue, making it difficult to control cell growth on the surface. In contrast, conducting polymers are emerging as they are relatively soft and adjustable, allowing modulation of the rigidity of cell cultures. In this section, we introduce the conducting materials that are commonly used for neural MPSs.

-Polypyrrole

Polypyrrole has emerged as a conducting material for neural tissue scaffolding in vitro. It is an organic polymerized pyrrole that promises biocompatibility [78] and is highly conductive when oxidized. It has been applied in various aspects of tissue engineering, including vessels, muscles, and neural tissue. Notably, a polypyrrole-based scaffold stabilizes neuron cell attachment and cellular outgrowth by adjusting the growth factors. Polypyrrole can not only upregulate the growth factors, but can also downregulate them. Zhang et al. [79] investigated the effect of a polypyrrole substrate on neuregulin-1 (NRG1) in a case of schizophrenia. The NRG1 modulated the neurite outgrowth, including the plasticity of the neural connection, and differentiation of neural stem cells. The dysregulation of NRG1 may induce schizophrenia, owing to the resultant weak neural plasticity. In this research, the normal neurons on the polypyrrole substrate showed improved neurite growth with electrical stimulation. Furthermore, the NRG1 knockout neurons also showed enhanced neural plasticity, along with normalized synaptophysin, PSD95, and brain-derived neurotrophic factor secretions related to synapse formation between neurons. Finally, they determined that a polypyrrole substrate can transport electrical stimulation, and that it could prevent failures of neurite outgrowth owing to NRG1 dysregulation. The results indicate that polypyrrole is a promising material for electrical stimulation in neural tissue engineering.

-Graphene

Graphene is also widely used conducting material. Graphene is a synthetic material made from carbon, and includes monolayers and honeycomb networks [80,81]. One of significant feature of graphene is high conductivity. High conductivity of graphene allows to build a platform to record and sense the electrical signal of neurons as well as induce electrical stimulation to neurons [80,82,83,84]. For example, Merhie et al. [85] functionalized MEA with single layer graphene (SLG) and recorded activity of neural networks. The neurons on the functionalized MEA with SLG showed higher survival rate and more synchronous behavior on the neural networks. The results indicate that the graphene is advantageous not only for electrical signal recording but also enhancing neural behavior.

Furthermore, one of the primary challenges for a neural MPS is to reconstruct a functional neural network. The functional neural network includes a systematic connection between each neuron. For that, a topographical cue for the ECM is essential for extending the axons out along with the structure. Graphene has attracted attention in neural tissue engineering for neural network reconstruction, owing to its topographical advantages as well as conducting characteristic. These features allow the graphene to provide topographical cues for guiding axon growth. Sakai et al. [86] developed a self-folding graphene/parylene- film to build a cell-encapsulated micro-roll (Figure 4). The porous graphene-based micro-roll made it possible to isolate the neuron body and axons in a 3D tubular structure for guiding neurite outgrowth. Moreover, the structure was expected to be able to reconstruct a brain-like neural network, by assembling blocks of guided neuron cell-laden micro-rolls.

Thus, it could allow researchers to connect engineered neural tissue to surrounding native tissue for regeneration, as well as for studying the therapeutic potentials of drugs by delivery via a porous structure. These experimental results are expected to provide insight for building a reliable in vitro neural tissue model (i.e., not only for neural tissue regeneration) based on the use of graphene.

In addition to the materials mentioned above, various biocompatible polymers have been applied in neural tissue engineering, such as polyaniline, polythiophene, and polyacetylene [87]. The combination of these conducting and biocompatible materials for neural tissue engineering is expected to build stable neural networks, via the electrical features of the materials. Furthermore, the conducting polymer can also combine with metals that have higher conductivity. As metals are too hard to adapt to neural cell growth, polymer coated metals or synthesized materials that mix metals and conducting polymers can also be used as probes for neural tissue stimulation with flexibility that does not induce mechanical damage to the neural tissue. For example, a flexible substrate coating that is easy to adapt to neural cells or the form of a nanocomposite, which is sufficiently adaptable to the ECM of neural tissue is applied [88]. In this manner, it is expected that the additional application of conducting polymers can complement the mechanical limitations of the existing metals.

### 3.2. Natural Source Derived Biomaterial

Even though synthetic materials are favorable for controlling mechanical properties such as the modulus, porosity, and degradability, synthetic materials have relatively less ability than native tissues to induce ECM remodeling for tissue reconstruction, and have different protein compositions relative to native tissues/organs. Hence, natural source derived biomaterials are focused on, when recreating an artificial scaffold that has a physiologically neural tissue-like environment because of their physical stiffness and composition of the materials [89,90]. Below, we address the popular natural materials for a favorable neural cell environment, and suggest approaches for each material to improve the favorable effects on the cells (Table 1).

#### 3.2.1. Hyaluronic Acid (HA)

HA is an anionic and nonsulfated glycosaminoglycan, and is abundant in the ECM of nervous tissue. It is practically advantageous for enhancing nerve regeneration, cell migration, and neuronal development [98]. Owing to these advantages, HA hydrogels are generally applied to in vitro neural tissue scaffolds. A HA for use in cellular purposes can be easily isolated from abundant natural resources, such as rooster combs, bovine eyes, and streptococcus qui. This accessibility has allowed for various applications of HA hydrogels to experimental designs [91,92,93].

However, one limitation in applying HA to a cell culture is that the cells cannot attach to the surface of the HA hydrogel itself without surface modification [94]. Therefore, strategies to blend HA hydrogels with other materials are widely used to enhance cell attachment. For example, poly-D-lysine, poly-L-lysine, and laminin are generally applied in HA hydrogel blending. In addition, a blended HA hydrogel can provide tunable mechanical properties that affect cellular behaviors. For example, Wu et al. [99] implemented methacrylated hyaluronic acid for an iPSC-neural progenitor cell (NPC) culture, and investigated the effect of the hydrogel rigidity on the differentiation and spheroid formation of the cell. They showed that using the iPSC-NPC with a soft hydrogel (approximately 0.5 kPa) allowed for more differentiated cells. Moreover, the neurite outgrowth and spheroid formation were enhanced with the soft conditions.

HA hydrogels have low batch-to-batch variation, and a stable structure. These features can provide an advantage over other natural source-derived materials in defining a composition. In addition, hydrogels can stably combine with other materials and have various applications, such as delivering neurotrophic factors, supporting neural structures, and providing a base material for organoid cultures.

#### 3.2.2. Matrigel

Matrigel is a commercialized ECM hydrogel derived from Engelbreth-Holm-Swarm mouse sarcoma cells [100]. The material is widely used for neural cell cultures, as well as organoid cultures. It contains laminin, collagen type IV, entactin, and various growth factors abundant in the brain; therefore, it is favorable for neural cell behaviors [100]. The laminin-rich property can fulfill an essential component of the brain. Laminin is one of the abundant proteins in the brain ECM; it promotes cell proliferation and differentiation, and supports neural cell attachment [101,102,103].

Owing to this favorable potential, Matrigel has contributed to building in vitro neural tissue models, and has been applied to NDs. Matrigel can be applied in engineer-controlled neural structure reconstruction. For example, Bang et al. [104] developed a cross-linked unidirectional Matrigel structure by inducing shear stress from a microfluid, and employed a primary neuron from rat cortex to build a neural circuit. The neural device showed a synaptic connection between a pre-synaptic neuron and post-synaptic neuron by engineering the ECM structure. The engineered matrix had favorable effects with regard to mimicking axon fasciculation and neural bundle formation.

Additionally, an ND model can be built on such a material for neural cell cultures. Choi et al. [41,105] produced a human neural progenitor cell line expressing the pathophysiology of AD, including amyloid-beta secretion and pathological tau formation. These studies successfully showed the advantages of Matrigel for neural cells, and that it was favorable for building neural structures.

Even though Matrigel has contributed to reconstructing in vitro neural tissue models and ND models, there is variance between each batch. Moreover, it cannot utterly recreate the protein composition of the native brain ECM. Furthermore, it is difficult to build a 3D structure, owing to its relatively low mechanical properties. Matrigel has high degradability, which is supportive for nutrient access and ECM remodeling; however, owing to the same degradability, it cannot maintain a hydrogel structure for a long-term neural cell cultures. For the further study of NDs, a long-term neural cell culture is necessary for reproducing aging factors by neural cell maturation. As such, additional components are needed to maintain the matrix structure for a longer time.

#### 3.2.3. Collagen

Collagen is an essential fibrous protein for maintaining the structures of tissues and organs in various organisms. Collagen can be easily isolated from animal tissues, such as pig, rat, and fish tissues. This makes it possible to apply collagen in various tissue modeling approaches. Collagen also is also easy to control experimentally, e.g., with shear stress and gravity [95].

Owing to these advantages, collagen has been used for neural tissue modeling. The brain also has a small amount of various collagens that form a fiber structure, which affects the brain stiffness. For example, Kim et al. [106] developed an anisotropically organized 3D device using rat tail collagen for reconstructing hippocampal neural networks. They fabricated a polymethylsiloxane (PDMS) mold and filled it with pre-crosslinked collagen hydrogel. Then, they induced a uniaxial strain to the mold, to provide residual stress for aligning the fibers in the collagen. The primary hippocampal neurons followed the anisotropically aligned fiber structure, and formed a neural circuit. The results indicated that collagen could be modified with a simple technique, and that the modified structure was favorable to neural circuit formation.

Even though collagen type I is often used to model the neural MPS, the material is not the protein primarily observed in the brain. Collagen type IV is considered the primary collagen component in the brain among the various types of proteins. Therefore, a collagen type I-based neural MPS can be controversial with regard to whether the neural cellular behavior in the neural MPS is “normal”. In addition, the mechanical properties of collagen are quite different from those of native brain proteins, including with regard to degradability, stiffness, and porosity. The physiologically different conditions may induce abnormal behaviors in neural cells. Thus, a clear rationale is required for evaluating neural cell behaviors on the material, such as by observing the reactivity of glial cells and axon outgrowth length for neuron differentiation.

#### 3.2.4. Decellularized Extracellular Matrix (dECM)

The decellularization of tissues and organs was first introduced in the tissue engineering field approximately a decade ago. The method removes most of cellular components and antigens in the native tissue, leaving the ECM. An ECM derived from decellularized tissues, known as a dECM, has emerged as an approach to recreating the physiological environment of native tissues in the view of its protein composition, growth factor, and physical properties [107]. Insofar as neural tissues, a brain is chiefly composed of an astonishing number of proteins, and each protein affects neural cell behaviors. For example, the brain mainly has glycosaminoglycans (such as laminin and HA), along with a small amount of collagens. The recapitulation of a protein composition with the engineered matrix has met difficulties, as the engineered matrix cannot reflect the complex protein composition. Therefore, the dECM has recently become attractive, as the dECM has the potential to preserve the original protein composition of the brain by adapting the native tissue. Therefore, the “brain dECM” has emerged as an approach to recapitulating a brain-specific microenvironment, particularly with regard to protein components and biocompatibility with neural cells [96,97].

The brain dECM has shown enhanced neurogenic effects regarding a physiologically mimicked microenvironment. Jin et al. [108] improved the yield of the direct reprogramming of fibroblasts into neurons in a brain dECM. They developed a human brain-derived ECM (BEM), which contained most of the ECM of the native brain. The directly reprogrammed fibroblasts expressed enhanced neuron-like characteristics, and provided a higher yield for non-viral reprogramming on a BEM-coated dish. In addition, animal experiments employed reprogrammed neurons (iN cell) on BEM dishes to improve neurobehavioral recovery in ischemic stroke animal models. The delivered iN cells contributed to the recovery of neurobehaviors. The results indicated that the recapitulation of the brain-specific microenvironment can enhance the expression of neuron differentiation.

Thus, the brain dECM is favorable to neural cell behavior enhancement. However, it requires several improvements. First, as noted above, batch-to-batch variation is a significant problem for a natural source-derived matrix. The differences are usually observed owing to the biological differences between each isolated brain. Accordingly, the field requires a standard for verifying batch-to-batch variation to measure the reproducible effect of a brain dECM on a neural cell. The second hurdle concerns the removal of an antigen in the brain dECM that induces immune reactions. The brain has primary immune cells called microglia that maintain the plasticity of the neural network. These cells quickly react to external stimuli. An animal-derived brain dECM has more antigens, which can trigger foreign body reactions. The removal of these antigens may facilitate the building of an immunity-free neural MPS with glial cells.

## 4. Manufacturing Method for Neural MPS Design

### 4.1. Soft Lithography

Soft lithography concerns a polymer casting process on a silicon master wafer fabricated by photolithography. After creating a solid mold with a micro-structure, an elastomeric polymer, normally PDMS, is cast on the mold to form a “chip”. The chip can be used to build technological components with precise control, e.g., microfluidic channels, micropumps, and electrode arrays (Figure 5A). The microfabricated platform can also be applied in tissue engineering, as the technical components facilitate mimicking the physiological interactions between cells, tissues, and organs. The chips, including the cellular components, are defined as an “organ-on-a-chip” [109]. The organ-on-a-chip is also used for neural MPSs, e.g., by reconstructing microenvironments of neural tissues and regulating neural cellular behaviors (including differentiation and outgrowth direction); furthermore, it allows researchers to reconstruct an environment physiologically. Moreover, the organ-on-a-chip has the potential to be adapted to pharmacokinetic models, as the model has high reproducibility [87].

As the technical components can simulate the interactions between cells, human-reliable disease conditions can be reconstructed on an organ-on-a-chip. Park et al. [62] designed a 3D microfluidic platform including AD gene expressed neurons, astrocytes, and microglia. They showed that the platform recreated the AD pathophysiology; moreover, it recreated the specific cellular behaviors of microglia activation regarding the ND pathophysiology. Microglia were placed on outside parts of the chip, and they penetrated the inside of the chip as they activated, owing to pathological factors. The chip compartmentalized the activated and inactivated microglia via microfluidic channels. The feature was strongly reconstructed in the model, providing a vital clue to the neuroinflammation owing to AD.

To mimic the complexity of NDs regarding progressive degeneration in the neural network, both the reproduction of ND condition and a recapture of the physiological structure are significant. One of the essential features of the brain is the aligned neural circuit. Microfluidic channels on an neural MPS can control neurite outgrowth, and form neural circuits [104,110]. Various approaches have been suggested for building a directional neural network with microfluidics. Na et al. [110] developed a neural diode with angle- and length-controlled microfluidic channels (Figure 5B). A diode in electrical circuits controlled the direction of electrical signal propagation, i.e., from anode to cathode, and not in the opposite direction. The neural circuit propagated the signal of the neurons from the pre-synaptic neurons to post-synaptic neurons as the diode. To construct the characteristics of the microfluidic channels, they determined that the neurite outgrowth could be suppressed above a certain angle of the channels. Based on this observation, they designed forward-bias channels and backward-bias channels via soft lithography on PDMS. The neurite bundles in the forward-bias channels could reach to the opposite side of the forward-bias population, but the bundles in the backward-bias channels could not. The neural diode was easy to fabricate, and provided good reproducibility with highly physiological features.

Neurite outgrowth can also be controlled via electrical stimulation. Honegger et al. [111] designed models with combined electric fields and microfluidics to control the neurite outgrowth direction. They showed that AC electrokinetic forces guided the neurite outgrowth speed and controlled the direction more precisely than when these actions were only microfluidic force-induced. Mechanical force can also be used to control the neurite outgrowth in a hydrogel scaffold. Kim et al. [106] showed how to fabricate an anisotropically arranged hydrogel scaffold to induce aligned neural networks. The collagen transited states based on thermal gelation. Before the collagen fully transited, the PDMS mold was stretched or compressed to provide anisotropic strain. The residual anisotropic strain induced the aligned fibrillogenesis of collagen, and make it possible for neurons to grow following the aligned fiber. This method was very simple fabrication process, and was effective for inducing an aligned neural network in vitro. They successfully reconstructed an ordered hippocampal CA3-CA1 circuit based on this method.

One of the specific physiological features of the brain is the BBB. The BBB is an endothelial border that selectively allows the penetration of external solutes and ions into the neural tissue. It has a tight junction between endothelial cells and pericytes. The tight junction makes selective semi-permeable barriers on the endothelium that protect the brain and maintain homeostasis in the brain [112,113,114]. The various factors of neurodegenerative disorders, such as neuroinflammation and gliosis, destroy the tight junction, and the BBB can no longer protect the brain from external materials and maintain optimal ionic composition [115,116,117]. Additionally, certain drugs cannot be delivered to the brain due to selective permeability. Therefore, recapitulation of the BBB is necessary to investigate the effect of neurodegeneration and drug delivery to the brain.

Hence, BBB on-a-chip has been developed in numerous ways. The BBB on-a-chip is one of the representative in vitro models to reproduce the BBB [118,119,120,121,122,123,124] (Figure 5C). Ahn et al. [50] developed a micro-engineered human BBB platform on a PDMS chip. The BBB platform successfully recapitulates complex BBB structures with precise alignment of the channels. Additionally, the human brain endothelial cell-based platform shows a high level of tight junctions, which is the main characteristic of BBB. Moreover, Ahn et al. proved that the model can be adapted to predict drug delivery potential into the brain by employing nanoparticles into the BBB models. In summary, the microfluidic-based BBB model can be employed in the quantitative analysis of the drug delivery into the brain for neurological diseases, such as AD and PD.

The research results show that chip fabrication is favorable for building various types of in vitro neural tissue. Organs-on-a-chip have a high degree of freedom with regard to fabrication and reproducibility. To obtain more physiological neural tissues on a chip, the organ-on-a-chip models require a more feasible neural cell source, i.e., one that expresses reliable characteristics and the pathologies of NDs. In addition, applying external components (such as electrode arrays and microfluidic components) is expected to facilitate obtaining more functional neural tissues-on-a-chip.

### 4.2. 3D Bioprinting

3D bioprinting is a biological application of additive manufacturing [125,126,127,128]. 3D structures can be created by conventional 3D printing though layer-by-layer buildup with various materials. Based on this principle, 3D bioprinting has been developed for tissue engineering, and it enables the reproduction of the structure of tissues and organs by precisely positioning them with a bio-ink. Bio-ink is an essential material for 3D bioprinting. It is a mixture of biocompatible materials, cells, and supporting chemicals. After preparing the proper bio-ink to reproduce the target tissue, various 3D printing methods can be adapted to build a tissue structure, such as extrusion printing with pneumatic pressure or mechanical screw plunger, inkjet printing with drop-by-drop patterning, and light-assisted printing that cure the structure with special light (laser or UV) (Figure 6A). The high freedom of fabrication in 3D bioprinting with various cells, materials, and methods can be advantageous for reconstructing complex tissue structures and organs [129].

3D bioprinting has also attracted attention for constructing a neural MPS, owing to advantages with regard to compartmentalization with patterning and comprehensive choices for selections of materials. The method is also favorable for building various shapes with different polymer and hydrogel scaffolds. The tunable characteristic of 3D cell-printing is also applicable to neural tissue engineering. Thus, 3D bioprinting has become an emerging fabrication method for reconstructing an neural MPS [125,127].

Even though 3D bioprinting has been widely used in tissue engineering, only a few cell-printed neural models are available, as the method was only recently introduced to in vitro neural modeling. Gu et al. [130] first suggested the direct-writing printing of neural stem cells, by encapsulating the cells in the biomaterial. They produced a bio-ink comprising polysaccharides alginate, carboxymethyl chitosan, and agarose. Then, neural stem cells were dispersed into the biopolymer ink and were directly printed to form a neural construct. The encapsulated neural stem cells showed sufficient viability, and maintained the pattern structure. The cells in the construct were principally differentiated into GABAergic neurons, together with glial cells. The results showed that 3D printing can form functional in vitro neural constructs, based on patterning (and on adapting the proper biomaterials).

3D printing strategies also make it possible to reproduce physiological features of the nervous system. Neurons form a highly connected and aligned neural network. Moreover, neural tissue has a heterogeneous environment composed of various cell types, including neurons and glial cells. Joung et al. [131] fabricated an engineered neural tissue that arranged neurons in the correct positions within a scaffold, via 3D bioprinting (Figure 6B). They first fabricated the biocompatible scaffold using layer-by-layer printing. They employed neural progenitor cells and oligodendrocyte progenitor cells for the 3D and heterogeneous cell layers, to form spinal cord tissue. They showed that the printed neural cells could rebuild functional axon connections, and could reproduce the spinal cord structure. Although the study aimed at regeneration following spinal cord injury, the strategy for reproducing the aligned axonal connections could be useful for studying the propagation mechanisms of NDs through the neural network.

Furthermore, 3D bioprinting has been developed for reconstruction of specific disease conditions, by controlling and patterning the dimensions of the structure. Yi et al. [132] developed cell-printed in vitro glioblastoma (GBM) models via extrusion bioprinting and a brain dECM. They designed pseudopalisading cells by patterning GBM cells and endothelium cells. The patterned structure showed a synergistic effect to expressing the pathophysiology of GBM, which was better than that of the non-patterned group. Moreover, the printed GBM-on-a-chip reproduced the treatment resistance in patient-specific GBM cells. The work showed that multimaterial and multicellular patterning with 3D bioprinting is favorable for reconstructing the pathologies of NDs.

To improve cell-printed in vitro neural tissues, bioprinting requires several technical advancements. First, any neuron-favorable biomaterials should be verified for their printability, physiological similarity, and enhanced effects for neural cellular behaviors. For example, Matrigel is a widely used material in neural tissue engineering; however, it has weak mechanical properties, not only for fabrication, but also for maintaining the structure. Furthermore, many printed neural tissues use collagen or silk owing to their controllable mechanical properties, even though these materials have significantly different physiological conditions from the native brain. In other words, brand-new biomaterials should be developed aiming to enhance the similarity with the native brain, and to improve the mechanical properties for fabrication. Second, the physical effects on neural cells during the printing process should be evaluated, e.g., the shear stress from extrusion nozzles and temperature effects. These physical effects may be harmful to neural cells; for example, glial cells can easily activate owing to these physical effects.

Moreover, building a physiologically relevant structure via printing is also important, as many of the current models are limited to building a lattice structure. The mimicry of the native brain structure, such as neural networks and compartmentalized structures, may be favorable to determining the propagation mechanisms of NDs. Taken together, 3D bioprinting has the potential to recreate the complex physiological structures in the CNS, and to adopt pathological factors.

## 5. Conclusions and Future Perspectives

Conventional experimental models, such as animal models and 2D cell culture models, have difficult hurdles to investigating the pathogenesis and therapeutic mechanism of an ND. The unmet needs concern, e.g., the genetic differences between human and animals, and the low physiological relevance to a patient’s pathophysiology. To address these unmet needs, neural MPSs have been used to obtain more physiological relevance with the CNS, by reconstructing the complicated characteristics of CNSs and NDs based on a highly delicate design strategy including cell sources, an artificial extracellular matrix, and fabrication methods.

The engineered neural cell sources have expressed stable phenotypes of neurons and pathogeneses of human ND conditions. Furthermore, the glial cell sources, such as astrocytes and microglia, have contributed to investigations of the astrogliosis and neuroinflammation that are pathologies of NDs. Moreover, various artificial ECMs, including those based on synthetic and natural source derived materials, have been favorable for mimicking the microenvironment of a CNS with the physiochemical features of each material. Lastly, the varied fabrication methods have made it possible to build physiological structures, such as neural networks and BBBs. The proper combination of cell sources, materials, and fabrication methods makes it possible to provide a physiologically relevant environment with a CNS.

To further improve the physiological relevance to the CNS and determine the complex pathologies of NDs, the neural MPS requires several improvements. First, including aging factors in the neural cell sources is expected to better reconstruct the pathological correlation between the neural MPS and in vivo conditions. The currently used cell sources (such as neonatal brain-derived neurons and stem cells) are controversial, as the age of the cells is younger than the patient’s chronological age [133]. Therefore, accelerating the cellular age can be one pharmaceutical strategy for determining the relationship between a drug exposure and restoration of degeneration.

Second, the artificial ECM requires improved physiological relevance with regard to its physical, chemical, and biological features. These features include a physical modulus similar to that of a native CNS, an ECM protein composition analogous to that of the native brain, and low neuroinflammatory factors that do not induce gliosis. Furthermore, the components of a natural source derived ECM should be clearly defined by improving the batch-to-batch variation, so as to investigate the exact effect of the ECM on the neural cell.

Finally, incorporating different fabrication methods can enhance the physiological relevance of the neural MPS to the CNS. An organ-on-a-chip based on soft lithography has advantages in adopting various mechanical architectures, such as micropumps, valves, and microfluidic channels. In addition, 3D printing is favorable for mimicking the inherent and heterogeneous interfaces of the CNS, based on the use of various bio-inks and high resolution of fabrication. Adapting these two technologies to the neural MPS together will make it possible to reconstruct a more complex structure of the CNS. In addition to the neural MPSs, organoids for neural tissue are also emerging for neurodegenerative disease since it can recapitulate complex structure of the neural tissue [134,135,136,137]. In this way, studying neurodegenerative disease in vitro using several technical methods will be a way to get closer to the solution for ND.

Current neural MPSs are limited to assessing specific effects of drugs on neural cells, and it is not possible to reproduce entire neurobehaviors. However, the neural MPS still has the pharmaceutical potential for examining drug toxicity and chemical safety directly on the CNS, as an artificial tissue [138,139]. Moreover, advances in the physiological relevance of neural MPS can be used to recreate interactions in the CNS that can highly affect the pharmacokinetic effects of the drugs in the CNS. Furthermore, the neural MPS might help to overcome the genetic differences from animal models and/or ethical issues regarding the models. In conclusion, the neural MPS can be used in a new alternative method for ND study and drug development.

## Figures and Tables

**Figure 1 micromachines-11-00855-f001:**
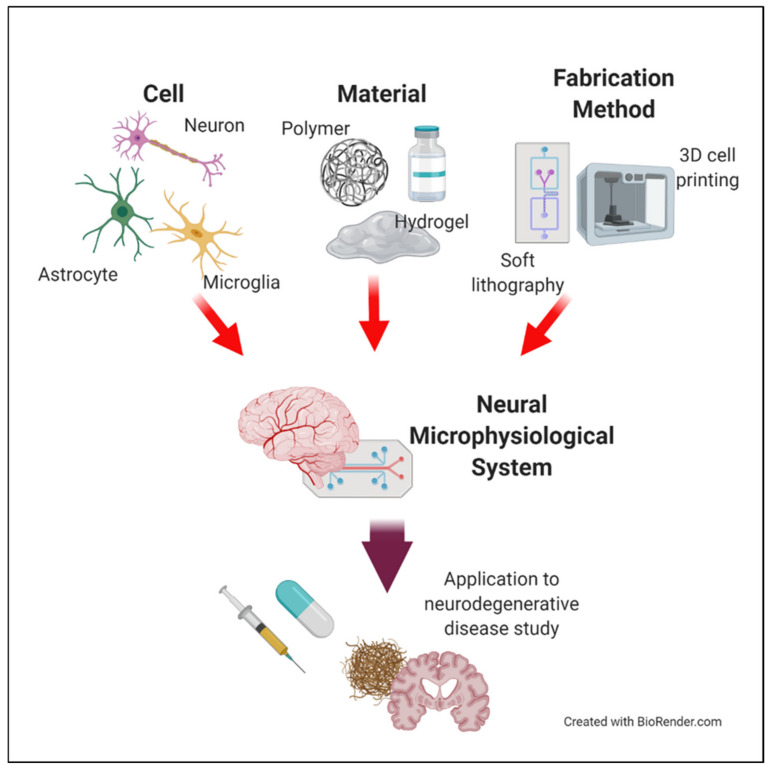
Schematic of the essential components for creating neural microphysiological system (MPS).

**Figure 2 micromachines-11-00855-f002:**
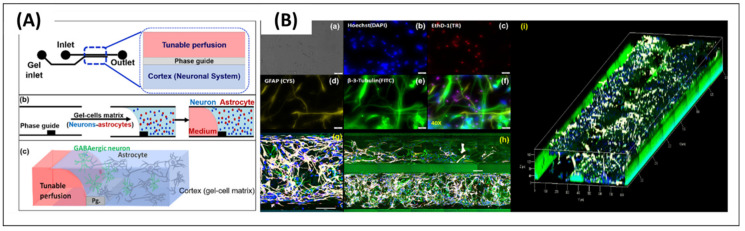
(**A**) Neurodegenerative disease modeling with various neuron cell sources; (**B**) Human induced pluripotent stem cell (iPSc)-derived neuron based neurodegenerative disease models. Reproduced with permission from Creative Commons Attribution [34].

**Figure 3 micromachines-11-00855-f003:**
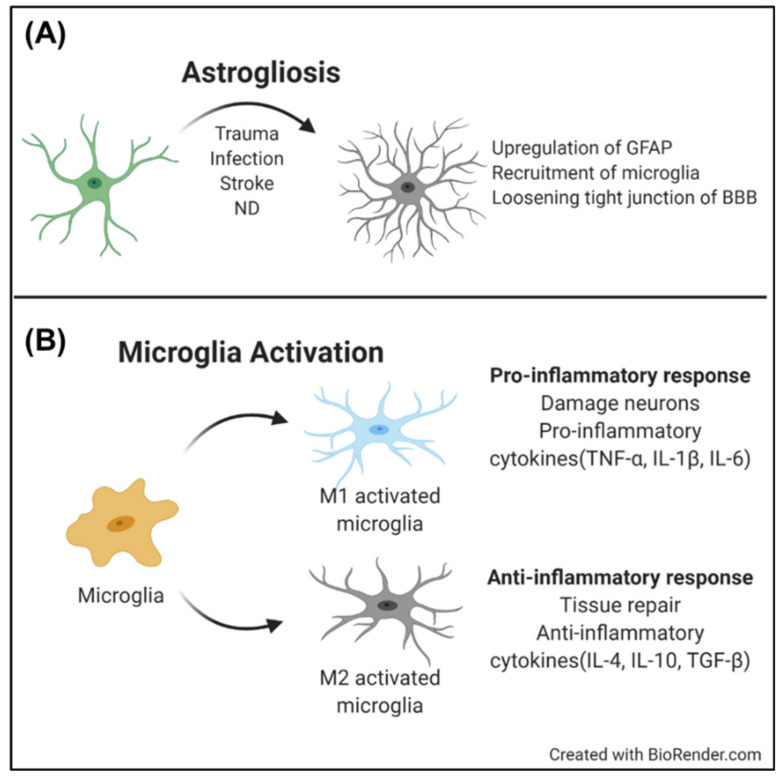
Behavior of glial cells upon activation (**A**) Gliosis of astrocytes; referred to [48] (**B**) Microglia activation; excessive pro-inflammatory response induces neuroinflammation in aged brains referred to [60].

**Figure 4 micromachines-11-00855-f004:**
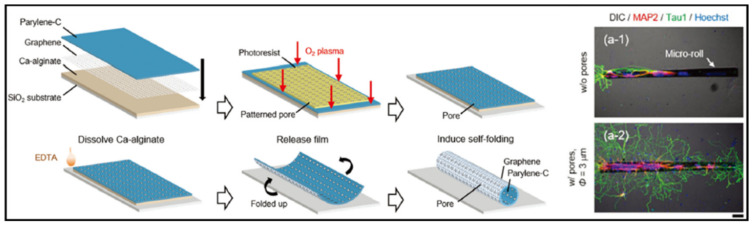
Application of conducting material to neural MPS; Graphene scaffold for neurite outgrowth. Reproduced with permission from [86].

**Figure 5 micromachines-11-00855-f005:**
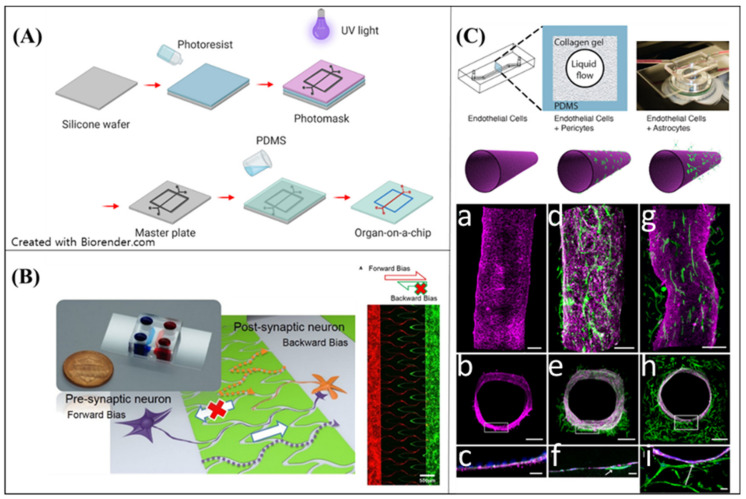
Soft lithography based neural MPS (**A**) Schematic of the soft-lithography progress for organ-on-a-chip (**B**) Neural diode on microfluid controlled model. Reproduced with permission from [110] (**C**) Microfluidic-based blood-brain barrier(BBB) model. Reproduced with permission from [124].

**Figure 6 micromachines-11-00855-f006:**
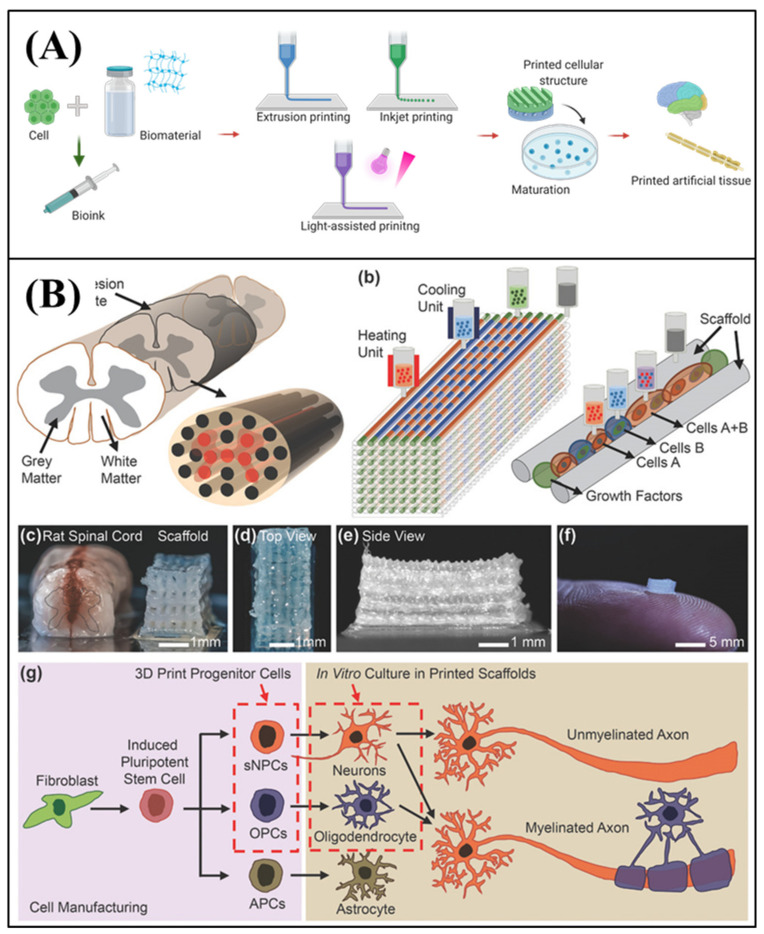
3D bioprinting based neural MPSs (**A**) Schematic of 3D bioprinting process for neural MPSs. (**B**) 3D cell-printed axon network model. Reproduced with permission from [131].

**Table 1 micromachines-11-00855-t001:** Natural source derived matrix for neural MPS.

Materials	Source	Advantages	Limitation
Hyaluronic acid (HA)	Rooster combs, bovine eyes, streptococcus qui [91,92,93].	High accessibility to isolate the material Low batch-to-batch variation Stable structure	Low cell attachment on the HA [94]
Matrigel	Engel-Holm-Swarm mouse sarcoma cells [94]	Various ECM proteins which are abundant in the CNS	Batch-to-batch variation Low mechanical property (degradability)
Collagen	Pig, rat, fish	High controllability for mechanical properties [95].	Lack of other critical proteins of CNS Different collagen type with main collagen in CNS
Decelluarized extracellular matrix(dECM)	Pig, human, mouse, etc.	Favorable to reproduce physiolological environment of the CNS (protein composition, physical properties) Enhancing neurogenic effects for neural cells [96,97].	Batch-to-batch variation Unrevealed protein composition

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
