# Peer review of "Microphysiological Systems for Neurodegenerative Diseases in Central Nervous System"

_micromachines, 2020, doi:10.3390/mi11090855_

Round 1

Reviewer 1 Report

The authors here present a review of recent progress toward artificial models of neural systems based on techniques like soft lithography and 3D printing. Overall, I felt the content was well written and the literature survey appeared complete.  However, tutorial reviews are intended in part to introduce the non-specialist to the field, and the survey here was not as approachable as it could be.  I have several suggestions below that would make the review more impactful; my comments are given below.

- Notably, I felt the paper would have benefitted from more (and more helpful) figures to help the reader, rather than relying so heavily on text.  Some of the figures that were included were not very useful (Fig. 1 in particular told me very little), and figures I would have expected to see (of the different types of cells, for instance, or giving visual demonstration of the two manufacturing approaches discussed) are minimal or absent. 

-In the materials section, I would have liked to have seen more comparing/contrasting of the materials approaches.  A summary table would be very helpful.

-the authors introduce the idea of using a conductive polymer like polypyrrole for electrical stimulation; polypyrrole is conductive, but does have orders of magnitude lower conductivity compared to conventional conductors like metals.  Does that pose a challenge here?  Is there an incentive to use a hybrid approach (a metal with a surface coating of polypyrrole, for instance)?

-In a tutorial review paper with a heavy medical component, I felt that it would have been useful to introduce the basics of 3D printing (what is 3D printing, defining the common terms used here (like ‘direct write’ printing), etc).  While the engineers among the audience may already be familiar with those terms, it would make the paper more useful to those in the biomedical space who may be less up to speed on manufacturing approaches.

Author Response

We thank the reviewer for their constructive comments and help in improving our manuscript. Please see the attachment for the response to the comments. 

Reviewer 2 Report

This is a nicely written review in the field of neuronal tissue engineering and organ-on-a-chip.

The text is easy to read and build up in a structured way. Writing is very good.

However I have some major aspects for revision:

1. The abbreviation MPNS is somehow unusual. I suggest to use better known words that describe the core of the review.

2. The review claims to summarize the state of art in the field of models for neurodegenerative diseases in general. This is an interesting goal, but is not exactly what I read. There are some sections that relate clearly to ND mainly in the cell chapter. However, I only found AD mentioned and discussed in several places. There are so many other ND that have not been mentioned at all, (except Schizophrenia briefly mentioned in l.343). I suggest to focus on AD as general statements in this field are difficult. If you want to have it for ND in general, you need to add a lot of more details, including a full literature research on this. 

That means you have to decide what story you want to tell: the state of art of microphysiological systems or models for ND.

3. In general, I find that there are not enough citations in the review. Some sections only take one or two articles in consideration. The state of art is not fully mirrored. This is an major issue for an review. 

4. I suggest to clearly separate in the text between scaffold material (like hydrogel, matrigel) and conducting materials (graphene, PP). 

5. You completely omitted (cortical) organoids which are an import new impact on neural models. Please add or comment on it.

6. Consider to describe systems for BBB in more detail.

Minor:

Fig 5C does not match with the text in l.570.

Author Response

(The authors gave the same response as above.)

Round 2

Reviewer 2 Report

minor corrections/additions:

l. 93-94 this sentence makes no sense to me rather delete.

l.199 As.... this sentence is weird, either delete or support with a citation.

Fig. 3 I dont find a reference nor any text to Fig 3a.

l.279 delete "in"

l.327 add a few citations

l.286. delete: ..., in the view ...materials. 

l.294 differentiation and aging ?, and ....    word missing.

l.360 Graphene is not a polymer !!!!!! see also figure 4

l.392 delete ]

l. 517 to 525 : rephase the section, the language is not precise nor is the content clear. I suggest to skip the  entire section. Mention the organoids in the conclusion with 1-2 sentences.

l.633 quotation mark missing

Author Response

Thank you for the kind and detail comment. We revised the manuscript according to the comment. Please refer to the attachment.
